# "*I got courage from knowing that even a daughter-in-law can earn her living*": Mixed methods evaluation of a family-centred intervention to prevent violence against women and girls in Nepal

**Nwabisa Shai**[1,2]*, **Geeta Devi Pradhan**[3], **Ratna Shrestha**[3], **Abhina Adhikari**[3], **Esnat Chirwa**[1,2], **Alice Kerr-Wilson**[4], **Rachel Jewkes**[1,2]

**1** Gender and Health Research Unit, South African Medical Research Council, Pretoria, South Africa, **2** School of Public Health, Faculty of Health Sciences, University of the Witwatersrand, Johannesburg, South Africa, **3** Voluntary Services Overseas (VSO) Nepal, Kathmandu, Nepal, **4** Social Development Direct, London, England, United Kingdom

* nwabisa.shai@mrc.ac.za

**Data Availability Statement:** All relevant data are within the manuscript and more detailed

## Abstract

### Background

We developed, and pilot tested a family focused intervention Sammanit Jeevan "Living with Dignity" to reduce gender-based violence by husbands, change harmful social and gender norms and improve the economic conditions of women through young married women-led income generating activities (IGAs).

### Methods

We conducted a modified interrupted time series study and qualitative research to evaluate the intervention in two migrant communities in Baglung district, Nepal. We enrolled young married women, their husbands and in-laws from 100 families. 200 women and 157 men completed questionnaires before the programme, and 6, 12 and 18 months afterwards. 18 in-depth interviews were conducted before the programme and 6 and 12 months later. We analysed the data for trends.

### Results

The intervention positively impacted young married women's economic conditions, exposure to violence and changed inequitable gender attitudes. Some positive outcomes were observed among older women and men. Young women's past month earnings (35.0% - 81.3%, β = 0.11, p-value<0.001) and savings (29.0% - 80.2%, β = 0.14, p-value<0.001) more than doubled over time. Young women experienced much less past year physical IPV over time (10% - 4.4%, β = -0.08, p-value = 0.077). They also perceived that their mothers-in-law were less cruel (mean 9.0–8.6, β = -0.03, p-value = 0.035). Improvements were observed in young women's individual (mean 44.4–43.3, β = -0.04, p-value = 0.297) and

questionnaires and data files are available on SAMRC MEDAT Data Repository and can be accessed with the following url: http://medat.samrc.ac.za/index.php/catalog/12.

**Funding:** This research was funded by UK aid from the UK government, via the What Works to Prevent Violence Against Women and Girls Global Programme. The funds were managed by the South African Medical Research Council. This brief draws on the findings from this original research; however, the views expressed do not necessarily reflect the UK government's official policies.

**Competing interests:** NO authors have competing interests.

perceived community gender attitudes (mean 54.4–51.4, β = -0.19, p-value<0.001) and they reported that their husbands were less controlling (mean 17.5–16.1, β = -007, p-value<0.001). These changes were supported by qualitative findings.

## Conclusions

Whilst caution is needed in attributing the effect due to lack of control arm, the results suggest that with adequate time and seed funding, Sammanit Jeevan enabled considerable income generation, a strengthened the position of young women in the households and it reduced their exposure to violence in this community. It warrants further research to optimise its impact.

## Background

Globally a third of women have experienced intimate partner violence (IPV) in their lifetime [1]. An estimated 37.7% of ever partner women from South-East Asia reported physical and/or sexual intimate partner violence in their lifetime and this is higher than other regions globally [2]. Of the 4120 women of reproductive age (15–49 years) who completed the National Demographic Health Survey (DHS) in Nepal, 32.4% had experienced emotional, physical and/sexual violence at the hands of their male intimate partners [3] and 28.3% reported these forms of violence in the past year [4].

The prevalence of IPV is highest among young married women, subject to violence by their husbands [5]. A cross-sectional survey that sampled 1296 married women aged 15 to 24 years found 51% had experienced some form of violence at the hands of their husbands while 46.5% reported sexual violence, 25.3% physical violence and 19.6% physical and sexual violence in Nepal [6]. Research in neighbouring India highlights the extent of IPV exposure for young married women as 62% of women who ever experienced marital violence did so in the first 2 years of marriage and married adolescents were twice as likely to be exposed to IPV compared to married adult women [7]. Being younger is an important risk indicator of married women's exposure to IPV, though older women continue to experience violence from their husbands [3, 8].

Women's exposure to violence can be explained by a confluence of individual, relationship, societal and political factors, most importantly driven by underlying patriarchal norms and values, that are highly prevalent [9]. Nepal is a patriarchal society and the family is founded on a patrilocal-patrilineal system that features social and gender based discrimination, privileging men over women, and socio–cultural norms and practices that legitimise the use of violence against women and girls [10, 11]. There are widely supported notions that men should be strong and dominant [12–14], positioned as primary decision-makers in marriage [12], protectors of the family who command respect and obedience, and exercisers of power controlling their women [15, 16]. Women are perceived as inferior, subordinate to men [12–14], and are expected to play a submissive and more conservative gender role when married, more especially in rural areas [16, 17]. Sons are much preferred over daughters, as they perpetuate the patriarchal family through marriage and care for their parents in old age [14, 18], while their sisters' destinies are believed to be tied to prospective affinal homes, where dowry is expected, despite its prohibition according to the Social Customs and Practices Act of 1976. Cultural practices impede women's agency and expose women and girls to violence, through the dowry system, child marriage [19], and the practice of "chhaupadi". The latter renders women impure

and untouchable during menstruation and restricts them from touching the family's property and food, including visiting other family and friends, or going to the temple or any festivals and events so as not to spread contamination [16]. There is a culture of silence on violence against women and girls in Nepal, shown by the low levels of help-seeking and reporting of incidents of violence in the home [20].

IPV is shaped in part by the intersection of gender inequality and poverty, which has far reaching consequences and increases women's social and economic dependence in relationships with men [21]. Nepal has a long-term history of labour migration dating back to the 19th century [22], and this is the main source of livelihood and employment for (mainly) men who remit funds to wives and parents whom they leave at home for their subsistence. 56% of Nepali households receive remittances at an average of 80,462 Nepali Rupees (~USD $800) per household annually [23], and remittances contributed 26.9% share of the GDP in 2016/2017 [24]. Though agricultural production in Nepal is largely dependent on the labour force of rural women [25], the long hours of women's labour are not recognised as productive. Lamichhane and colleagues assert that women in low occupations are more vulnerable to IPV than women in service or small business [6]. About three in four women (74%) are economically dependent on their husband [26] and few women participate in household decision making and fewer have control over household earnings [27]. Analysis of the national DHS shows that poorer women have a higher prevalence of IPV than wealthier women in Nepal. Women from more disadvantaged neighbourhoods are more vulnerable to abuse by male intimate partners compared to women from less disadvantaged neighbourhoods [3]. Women and girls have limited access to education and employment compared to men and boys. Their lower educational attainment, higher unemployment, and lower economic status increase their vulnerability to violence and dependence on their husbands [3, 4, 8, 28].

Family level factors such as witnessing domestic violence in childhood heighten women's likelihood of experiencing violence in intimate relationships [3, 4]. In marriage, women who report that their husbands are controlling are more often exposed to IPV [3, 29], as are those who fear their husbands [3]. Women who report their husbands' problematic use of alcohol are at higher risk of experiencing IPV [4, 29].

In a contemporary rural Nepali setting, unequal gender power relations within the family play a significant role in shaping young married women's IPV risk, from the imbalances in gender power between husbands and wives, and between the mother-in-law and daughter-in-law. While violence against young married women may be perpetrated by any member of the husbands' family, including the father-in-law, the brother-in-law or sister-in-law, research shows that the mother-in-law is the most common instigator of IPV and perpetrator of domestic violence against the daughter-in-law [16, 30].

Many women marry young, often before the minimum legal age of 20 years (the Marriage Registration Act) and this results in young married women in particular having limited prospects of further education [31], and less access to property and employment opportunities [32]. Husbands of young wives are more likely to enlist for migrant labour outside the country leaving their young wives under the authority of their families and with limited resources [33]. By custom, the mother-in-law assumes seniority over the daughter-in-law, takes charge of the orientation of the daughter-in-law into gendered roles within the affinal home including household chores, regulates her relations and mobility, and controls her access to food and her migrant husband's remittances [16].

Baseline findings from the One Community One Family project evaluation show that being older, have controlling husbands, having poor relations with their husband, and having to borrow money or food, and holding inequitable gender attitudes, increased all women's exposure to IPV while having a kinder mother-in-law was protective [28]. This provided a strong

indication that an intervention designed like Sammanit Jeevan may be impactful in this setting. The overall context of IPV in Nepal demonstrates the need to employ a holistic approach to address the social and structural factors increasing young married women's risk of experiencing violence. Such a programme would want to focus on preventing VAWG through changing unequal gender relations between young married women and their husbands and in-laws, as well as improving their economic conditions in ways that positively enhance family relations. Scholars promoting women's economic strengthening interventions suggest the family approach can be valuable in mitigating unequal gender relations [34, 35]. As we implemented the One Community One Family project we sought to adapt and test the effectiveness of Sammanit Jeevan (Living with Dignity)–a family focused intervention–that was designed to reduce gender-based violence by husbands, change harmful social and gender norms and improve the economic conditions of young married women living in two rural communities in Nepal. This evaluation was conducted by the South African Medical Research Council (SAMRC), Voluntary Services Overseas (VSO) Nepal and Bhimapokhara Youth Club (BYC) from 2016 to 2018 as part of the DFID-funded What Works to Prevent Violence Against Women and Girls? Global Programme.

## Methods

### Study design

The study used qualitative and quantitative methods to assess the effectiveness of the Sammanit Jeevan intervention and sought to fit together the insights provided by both research methods to better understand the relationship between the research outcomes and the intervention we delivered [36]. The quantitative study was a modified interrupted time series with four data points: baseline pre-intervention, 6 months post baseline, 12 months post baseline and 18 months post baseline. The baseline was done in January-February 2017 and the endline in August 2018. There was no control arm. Qualitative interviews were conducted at baseline, and 6 and 12 months after baseline.

### Participants

The sample size was calculated using young married women's experience of IPV as the primary outcome. We used the Cochran-Armitage trend test in Stata 13 [37] to calculate the minimum sample size and assumed 1 in 3 young women would have experienced some form of intimate partner violence in the 12 months preceding the intervention. It was anticipated by 18 months post baseline, about 15% young women would report experiencing IPV. The minimum sample of young married women required at 80% power and 5% significant level was 89. In total 100 families were selected with the intention of targeting 200 women (100 young & 100 older) and 200 (100 young & 100 older men).

At baseline, 100 families were recruited through a random allocation from household lists in each Village Development Committees (VDCs). Families were included if a young married woman was willing to participate in the study; her husband and in-laws were enrolled thereafter if they consented. For each family, one young married woman, her mother-in-law and where possible, her husband and her father-in-law participated in the intervention and the associated research. 30% of young married women lived just with their husbands and children, and 69% of them lived with their husbands and in-law. Husbands were accessed for enrolment into the study as they were home and may or may not have been migrating for work at the time. Fig 1 shows a flow chart of the number of participants interviewed during the course of the study, and young married women are described as 'young women', mothers-in-law as 'older women', husbands of young married women as 'young men' and fathers-in-law are

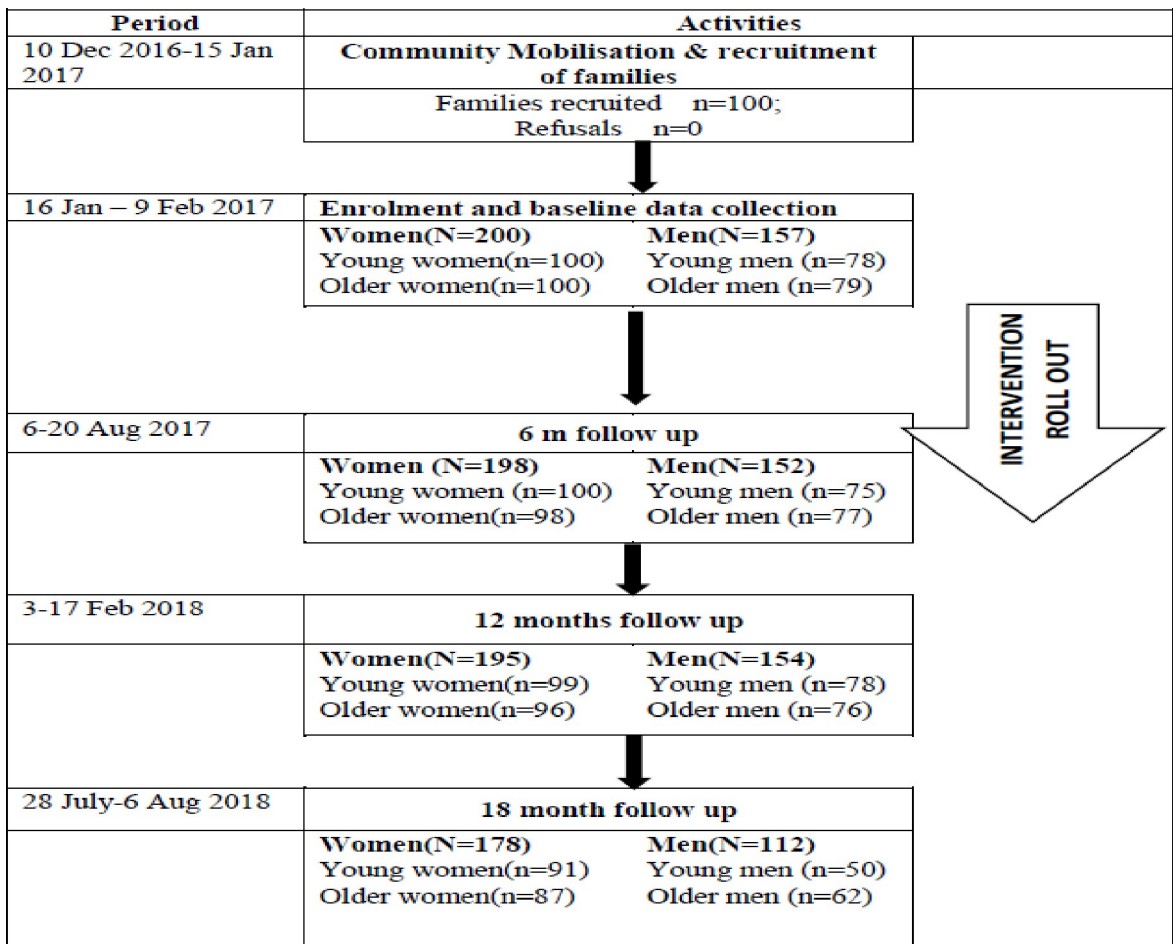

| Period | Activities | |
|---|---|---|
| 10 Dec 2016–15 Jan 2017 | **Community Mobilisation & recruitment of families** | |
| | Families recruited    n=100; Refusals    n=0 | |
| 16 Jan – 9 Feb 2017 | **Enrolment and baseline data collection** | |
| | **Women(N=200)** Young women(n=100) Older women(n=100) | **Men(N=157)** Young men (n=78) Older men (n=79) |
| 6-20 Aug 2017 | **6 m follow up** | |
| | **Women (N=198)** Young women (n=100) Older women(n=98) | **Men(N=152)** Young men (n=75) Older men (n=77) |
| 3-17 Feb 2018 | **12 months follow up** | |
| | **Women(N=195)** Young women(n=99) Older women(n=96) | **Men(N=154)** Young men (n=78) Older men (n=76) |
| 28 July-6 Aug 2018 | **18 month follow up** | |
| | **Women(N=178)** Young women(n=91) Older women(n=87) | **Men(N=112)** Young men (n=50) Older men (n=62) |

INTERVENTION ROLL OUT

**Fig 1. Flowchart of participants in the study.**

recorded as 'older men'. Fewer men than women were recruited due to the prevalence of male labour migration. The response rate reduced over time to the final data point where interviews were conducted with 89% of women and 71% of men. The qualitative interviews held with 6 young married women, 4 older women, 4 young married men and 2 older men, and most completed the 6 months and 12 months interviews.

A local non-governmental organisation, Bhimapokhara Youth Club, recruited participants after obtaining permission for the study from district level stakeholders and community leaders. Participants were eligible if they were agreed to participate in the intervention and the research, were mentally competent to give informed consent and if they could communicate in Nepalese.

## Study setting

The study took place in two of the 59 VDCs in rural Baglung district, Nepal. The main feature of Baglung district is its hilly landscape and it has been named the 'district of suspension bridges' owing to the numerous suspension bridges connecting the majority of the population settled on the river valleys around the area. Much of the population practices subsistence farming due to the fertile plains situated on the riversides, however the district is prone to landslides and flooding during monsoon season. Baglung also has one of the highest levels of overseas

male migration for labour. Like most of Nepal, the district has diverse religions, cultures and ethnic groups with Hinduism and Buddhism being the major religions, while Brahman, Chhetri, Janajati, and Dalit are among the ethnic groups and castes living in the area. The two VDCs covered nine municipal wards and six of these were selected for the research.

## The intervention

Sammanit Jeevan is a participatory, group based, and family-centred model to reduce intimate partner violence (IPV), change harmful gender and social norms and improve young married women's economic conditions through women-led income generating activities (IGAs) [38]. The programme intergenerationally has a workshop series that combines three elements: gender transformative norms, economic empowerment, and IGAs support. These elements are implemented in succession: first, the 10 gender transformative norms sessions set the tone to build better communication and more gender equitable and harmonious relations between young married women and their husbands and in-laws, second, the 3 economic empowerment sessions aimed at creating appreciation of the value of women's work in the home to facilitate the successful implementation of the third element. The third element had 7 sessions focused on building skills for and supporting IGAs led by young married women. It helps families to identify an IGA they are comfortable to run and provides the necessary business skills such as market analysis, business planning, budgeting, and each family receives in-kind start-up funding equivalent to $150, this was given in materials or livestock to start up the IGA. The model of the programme ensured young married women led their IGA, with assistance from at least one family member, and chose IGAs most familiar to them and suitable for local conditions.

The combined programme had 20 three-hour sessions delivered by trained facilitators from a similar background to the participants. One male and one female facilitator paired up to deliver the intervention to once weekly with separate same age-sex groups, that is 20 young married women, 20 older women and 15 young married men and 15 older men, and the groups came together for discussions after every three sessions. Facilitator pairs were responsible for 2 groups each. Intervention implementation took place over 20 weeks to deliver between mid-February and end August 2017.

The programme was adapted from the Zindagii Shoista model developed in Tajikistan [39, 40], which drew on Stepping Stones and Creating Futures interventions from South Africa [41, 42] and was evaluated under the What Works to Prevent VAWG? global initiative [39]. A participatory workshop, review, and pilot testing established the relevance and appropriateness of the adapted manual to the Nepali social and economic context. The Nepali adaptation also addresses the discriminatory traditional practices related to women's sexual and reproductive health needs and rights, and the types of IGAs commonly run by families in rural migrant communities.

## Data collection

All participants in the quantitative research were interviewed using a standard questionnaire administered by a trained interviewer. Interviewers were paired to participants by gender. The questions covered the socio-economic situation of respondents, family relations, gender attitudes, men's perpetration of intimate partner violence and women's experiences of it, physical and mental health, as well as hopes for the future. The survey used mostly the same questionnaires as that used in Tajikistan and drew on measures previously applied in other studies (see Table 1 below).

In-depth interviews were conducted with a purposive sample of young married women, their mothers-in-law, husbands and fathers-in-law. The interviews explored participants'

**Table 1. Primary and secondary outcomes.**

| Construct | Measure | Typical item | Origin |
|---|---|---|---|
| **Primary outcomes** | | | |
| Physical IPV in past year | 5 items to measure experience (women) and perpetration (men) of any act 1 or more times | (men) In the past 12 months, how many times have you slapped your current wife or thrown something at her that could hurt her? (women) In the past 12 months, how many times has your current husband slapped you or thrown something at you that could hurt you? | Garcia-Moreno et al (2006) |
| Sexual IPV in past year | 3 items to measure experience (women) and perpetration (men) of any act 1 or more times | (men) In the past 12 months, how many times have you physically forced your current wife to have sex when she did not want to? (women) In the past 12 months, how many times has your current husband physically forced you to have sex when you did not want to? | |
| Emotional IPV in past year | 7 items to measure experience (women) and perpetration (men) of any act 1 or more times | (men) In the past 12 months, how often did you stop your current wife from getting a job, going to work, trading or earning money? (women) In the past 12 months, how often has your current husband stopped you getting a job, going to work, trading or earning money? | |
| Any earnings in past month | Single item measure of whether participants have earned money in past month | Considering all the money you earned from jobs or selling things, how much did you earn last month? | |
| Any savings in past month | Single item measure of whether participants have any saved money in past month | How many Nepal Rupees did you put aside for savings in the last 4 weeks? | |
| Overall savings | Single item measure of whether participants have any savings | How many Nepal Rupees have you got in savings? | |
| **Secondary outcomes** | | | |
| Food insecurity | Household members' experience of a lack of food in the previous 4 weeks (3 items) | In the past 4 weeks how often did you or any member of your household go to sleep hungry because of lack of food? | Coates et al (2007) Household Food Insecurity Access Scale (HFIAS) |
| Stress due to unemployment | Stress due to current work (4 items) | I am frequently stressed or depressed because of not having enough work. | Barker G, et a. (2011) Images study questionnaire |
| Income seeking effort scale | Effort in getting job or earn income in the past 3 months (7 items) | How often have you in the last 3 months. . . handed in or sent off an application for work? | |
| Unemployment shame | Measure of shame or despondency due to unemployment (4 items) | I sometimes feel ashamed to face my family because I am out of work. | |
| Difficulty in borrowing money | Single item measure difficulty in borrowing money in emergency | If you had an emergency at home and needed 1500 Nepal Rupees how easy would you say it would be to find the money? | |
| Financial strain (borrowed food or money) | Single item measure in borrowing food or money | How often in the past 4 weeks have you had to borrow food or money because you did not have enough? | |
| Depression | CES-D scale (used as a continuous score) | During the past week I was bothered by things that usually don't bother me | Radloff (1977) |
| Life satisfaction | 4 item scale to measure participants satisfaction with life | In most ways my life is close to my ideal | |
| Suicidal thoughts | Single item on suicidal thoughts in the last 4 weeks | In the past 4 weeks, has the thought of ending your life been in your mind? | |
| Self-rated health | Single item with 5 categories to assess wellbeing | In general, would you describe your overall health as excellent, good, fair, poor or very poor? | |
| Disability | 6 items to assess disability due to health problem or injury | Do you have difficulty seeing even if wearing glasses? | Washington Group scale |
| Individual gender attitudes | 22 items to measure participants' gender inequitable attitudes | I think the wives in my family must ask permission from their husbands or husbands' family before going somewhere | Jewkes et al (2003) lightly adapted for Nepalese context after formative research |
| Community gender attitudes | 22 items to measure community gender inequitable attitudes | In my community many people think that a wife must ask permission from her husband or his family before going somewhere | |

*(Continued)*

**Table 1.** (Continued)

| Construct | Measure | Typical item | Origin |
|---|---|---|---|
| Wife's relationship with her husband | 7 items to measure wife's relationship with husband | My husband is very strict and controlling. | Based on Jewkes et al (2011) adapted for the Nepalese context |
| Husband's relationship with his wife | 7 items to measure relationship with wife | My wife does not really understand me. | |
| Women's mother-in-law kindness | 3 items to measure mother-in-law kindness, scored high = more kind | My mother-in-law loves me like her own daughter. | |
| Women's mother-in-law cruelty | 4 items to measure mother-in-law cruelty, scored high = more cruel | My mother-in-law is very strict and controlling. | |
| Man's assessment of his mother's kindness | 3 items to measure kindness of mother towards son, scored high = more kind | My mother is a kind person. | |
| Man's assessment of his mother's cruelty | 4 items to measure cruelty of mother towards son, scored high = more cruel | My mother can be cruel. | |
| Woman's involvement in decision making in the home | 5 items on woman's involvement in decision making, scored high = more involved | In the last 3 months, have your views been listened to on matters concerning the children and their schooling or work in your home? | |
| Relationship control | 8 items to measure man's controlling behaviour | (woman) He won't let me wear certain things. (man) I won't let her wear certain things. | Based on Jewkes et al (2008) adapted for the Nepalese context after formative research |

background, the circumstances surrounding the young women's marriage, the nature of gender roles and power relations and violence in the home, as well as household decision making, access to income including young married women's involvement in the income generating activities after participating in the Sammanit Jeevan intervention.

Business assistants from BYC conducted monthly monitoring visits to families engaged in IGAs and collected routine data on the implementation of IGAs per family including the type of IGA each family chose, obtained confirmation that the family IGAs were led by the young married women supported by their families, the income and savings earned over time, as well as the amounts of money the families invested. Monitoring visits also enabled the business assistants to troubleshoot, impart further skills, and support families to maintain their IGAs.

## Data analysis

Analysis was by intention to treat (ITT), thus we included all participants enrolled into the intervention at baseline irrespective of their attendance in the intervention training programmes. Descriptive statistics such as frequencies, percentages, means and standard deviations were used to summarise socio-demographic characteristics of participants. We used percentages to summarise categorical or dichotomous outcomes. For scales such as gender attitudes and decision-making scales, we derived an additive score from the item responses and presented mean scores as summary statistic at each time point. Prior to testing for effect of intervention, we performed some missing data analysis to check for any association between loss to follow up (not being available at 18m) and intervention outcomes at baseline. As expected due work migration, the proportion of younger men not available at 18 m was higher than that of older men. However, we found no association between loss to follow up and any of the intervention outcomes at baseline. Thus, no imputation for missing outcomes was done for participants not available at 18 months.

Generalised random effects regression models were used to assess the trends in proportions or mean score over time, with each participant and their set of interview data as a random component (cluster) in the model. Study time points were entered as the main exposure and

were used to determine trend in each outcome measure over time. All random effects models adjusted for the age of the participants and all analyses were done in Stata 15. We analysed the data for male and female participants separately.

The mixed methods analysis used in this paper seeks to explore complementarity between the quantitative results and qualitative findings, to obtain understanding of the context, illustrate and clarify our quantitative results using the qualitative research findings [36]. We undertook thematic analysis of the transcribed interviews to generate insights and interpret data on the nature of gender and power relations within families and young married women's participation in income generation and to interpret the conceptualizations informing participants' articulations [43, 44]. The research team worked together in developing inductive thematic codes from the data and interpreted participants' experiences and perceptions within the context of rural Nepal. We followed six phases of thematic analysis: familiarised ourselves with the data by reading the transcribed interviews, systematically generated codes, search for, reviewed, defined and named the emerging themes [43], based on a codebook for which we established intercoder reliability [45]. We then summarised the key themes: relations with spouse and in-laws, gender attitudes and roles, women's experiences of and men's perpetration of IPV (and women's domestic violence from in-laws), access to income, decision making and young married women's participation in IGAs, to corroborate and contract with the quantitative results from the surveys.

Data collected on the IGA implementation was recorded in spreadsheets and analysed per type of IGA. This included the families' earnings and savings over time, and maintenance of IGAs.

### Ethical considerations

We received ethics approval from the South African Medical Research Council and the Nepal Heath Research Council for this study. We explained the study as a project seeking to understand health, family life and relationship and the intervention was described as a programme designed to strengthen relationships and livelihoods. Participants signed informed consent before interviews where the purpose, methods and key details of the study were explained in all rounds of data collection, and participants allowed to ask questions and responses to those questions were provided. Participants were assigned a unique identification code which was stored on the questionnaire and in-depth interviews. Each interview was conducted in private to ensure that interviews could not be overheard by others and confidentiality was assured. Young married women were interviewed on different days to their husbands and in-laws and vice versa, and participants were encouraged not to discuss the interviews as everyone would have their turn, and their stories may differ. None of the young married women report adverse reactions of husbands and in-law after the interviews. Participants were given information on available VAWG related psychosocial, legal and medical services available in the community and the district. A packet of seasonal vegetable seeds was provided to each participant to compensate for their time.

### Results

Of the 200 women enrolled into the quantitative surveys at baseline, 178 (89%) were available for interviews at all the 4 data waves, and 27 (8.5%) were available for the baseline, 6 months and 12 months interviews. Only 5 women were not available for the 12 months' and 18 months' data collection. Not being available for the 18 months data collection was neither associated with baseline IPV experience nor with being young or older woman. Of the 157 men enrolled for the quantitative study at baseline, 112 (71%) were available for interviews at all the

**Table 2. Baseline socio-demographic factors and exposure to IPV among women and men and among younger and older women.**

| | Women | | | | | | Men | | | | | |
|---|---|---|---|---|---|---|---|---|---|---|---|---|
| | Young(n = 100) | | Old (n = 100) | | All (n = 200) | | Young(n = 78) | | Old (n = 79) | | All (n = 157) | |
| | n | % | n | % | n | % | n | % | n | % | n | % |
| **Age group** | | | | | | | | | | | | |
| 16-24yrs | 47 | 47.0 | 0 | 0 | 47 | 23.5 | 16 | 20.5 | 0 | 0 | 16 | 10.2 |
| 25-34yrs | 29 | 29.0 | 1 | 1.0 | 30 | 15.0 | 39 | 50.0 | 1 | 1.3 | 40 | 25.5 |
| 35-44yrs | 23 | 23.0 | 8 | 8.0 | 31 | 15.5 | 17 | 21.8 | 4 | 5.1 | 21 | 13.4 |
| 45-54yrs | 1 | 1.0 | 30 | 30.0 | 31 | 15.5 | 6 | 7.7 | 16 | 20.3 | 22 | 14.0 |
| > = 55yrs | 0 | 0 | 61 | 61.0 | 61 | 30.5 | 0 | 0 | 58 | 73.4 | 58 | 36.9 |
| **Ethnicity** | | | | | | | | | | | | |
| Dalit | 14 | 14.0 | 13 | 13.0 | 27 | 13.5 | 9 | 11.5 | 9 | 11.4 | 18 | 11.5 |
| Janajati | 44 | 44.0 | 45 | 45.0 | 89 | 44.5 | 35 | 44.9 | 36 | 45.6 | 71 | 45.2 |
| Chhetri | 29 | 29.0 | 27 | 27.0 | 56 | 28.0 | 25 | 32.1 | 23 | 29.1 | 48 | 30.6 |
| Brahmin | 12 | 12.0 | 13 | 13.0 | 25 | 12.5 | 9 | 11.5 | 10 | 12.7 | 19 | 12.1 |
| Other | 1 | 1.0 | 2 | 2.0 | 3 | 1.5 | 0 | 0 | 1 | 1.3 | 1 | 0.6 |
| **Education level** | | | | | | | | | | | | |
| None | 14 | 14.0 | 92 | 92.0 | 106 | 53.0 | 1 | 1.3 | 40 | 50.6 | 41 | 26.1 |
| Primary | 11 | 11.0 | 7 | 7.0 | 18 | 9.0 | 7 | 9.0 | 25 | 31.7 | 32 | 20.4 |
| Secondary | 53 | 53.0 | 0 | 0 | 53 | 26.5 | 44 | 56.4 | 11 | 13.9 | 55 | 35.0 |
| Above SEC | 22 | 22.0 | 1 | 1.0 | 23 | 11.5 | 26 | 33.3 | 3 | 3.8 | 29 | 18.5 |
| **Currently married** | 99 | 99.0 | 87 | 87 | 186 | 93 | 74 | 94.9 | 77 | 97.5 | 151 | 96.2 |
| **In polygamy** | 3 | 3.0 | 28 | 28 | 31 | 15.5 | 0 | 0 | 7 | 8.9 | 7 | 4.6 |
| **Married to a relative** | 26 | 26.0 | 19 | 19 | 45 | 22.5 | 13 | 16.7 | 16 | 20.3 | 29 | 18.5 |
| **Migrated for work** | 1 | 1.0 | 0 | 0 | 1 | 0.5 | 56 | 71.8 | 52 | 65.8 | 108 | 68.8 |
| **Lifetime IPV experienced/perpetrated:** | | | | | | | | | | | | |
| **Physical IPV** | 20 | 20.0 | 24 | 24.2 | 44 | 22.1 | 12 | 16.0 | 16 | 20.3 | 28 | 18.2 |
| **Sexual IPV** | 7 | 7.0 | 20 | 20.2 | 27 | 13.6 | 0 | 0 | 1 | 1.3 | 1 | 0.6 |
| **Sexual or physical IPV** | 24 | 24.0 | 33 | 33.3 | 57 | 28.6 | 12 | 16.0 | 16 | 20.2 | 28 | 18.2 |
| **Past year IPV experience/perpetration:** | | | | | | | | | | | | |
| **Physical IPV** | 10 | 10.0 | 6 | 6.1 | 16 | 8.0 | 3 | 4.0 | 4 | 5.1 | 7 | 4.6 |
| **Sexual IPV** | 4 | 4.0 | 6 | 6.1 | 10 | 5.0 | 0 | 0 | 1 | 1.3 | 1 | 07 |
| **Sexual or Physical** | 13 | 13.0 | 9 | 9.1 | 22 | 11.1 | 3 | 4.0 | 4 | 5.1 | 7 | 4.6 |
| **Any IPV** | 23 | 23.0 | 17 | 17.2 | 40 | 20.1 | 5 | 6.7 | 6 | 7.6 | 11 | 7.1 |

4 data waves, and a quarter (39/157) were available at baseline, 6 months and 12 months. About a third (35.9%) of the young men were not available for the 18 months data collection compared to 21.5% of the older men. However, not being available at 18 months was not associated with baseline violence perpetration. There were 11 deaths of study participants during the study: 9 were from natural causes, 1 died from a car accident during the monsoon season and 1 reportedly committed suicide at his mother's funeral. All deaths and adverse events were investigated, and none was linked to activities in the study.

Table 2 shows the baseline socio-demographic characteristics and exposure to IPV among the women and men enrolled in the study. Over half of women participants were aged 16 to 35 years and over a third of the men were in the same age group. Most participants came from the Janajati and Chhetri ethnic groups. 53% of women had no education compared to 26.1% of men. Many of the participants were currently married, 15.5% of women and 4.5% of men were in polygamous marriages while a few women and men had married a relative. Migration was much more common for men than women. 28.5% of women had experienced physical

**Table 3. Income, work and household insecurities among women and men, and among young versus older women.**

| | All women | | | | | | All men | | | | | |
|---|---|---|---|---|---|---|---|---|---|---|---|---|
| | baseline | 6m | 12m | 18m | | | baseline | 6m | 12m | 18m | | |
| | mean /% | mean /% | mean /% | mean/ % | coef | p-value | mean/ % | mean/ % | mean/ % | mean/ % | coef | p-value |
| | n = 200 | n = 198 | n = 195 | n = 178 | | | n = 157 | n = 152 | n = 154 | n = 112 | | |
| Have earnings in past month (%) | 21.0 | 22.2 | 12.8 | 64.6 | 0.14 | <0.001 | 52.9 | 53.3 | 64.3 | 75.0 | 0.12 | <0.001 |
| Any savings in past month (%) | 17.0 | 18.2 | 15.9 | 63.5 | 0.17 | <0.001 | 27.4 | 33.6 | 54.6 | 50.9 | 0.15 | <0.001 |
| Any savings at all (%) | 51.5 | 56.6 | 67.7 | 96.6 | 0.19 | <0.001 | 36.3 | 47.4 | 60.4 | 83 | 0.21 | <0.001 |
| Work stress score (high = more effort) | 8.8 | 8.3 | 7.8 | 9.8 | 0.04 | <0.001 | 10.0 | 9.4 | 9.0 | 11 | 0.03 | 0.044 |
| Unemployment stress (high = more stress) | 9.7 | 9.4 | 9.1 | 10.1 | 0.01 | 0.165 | 9.3 | 9.2 | 9.9 | 9.0 | 0.01 | 0.534 |
| Work shame (high = more shame) | 9.1 | 8.5 | 8.2 | 8.2 | -0.05 | <0.001 | 8.7 | 8.3 | 8.9 | 7.9 | -0.03 | 0.019 |
| Done anything to earn money past 3 months (%) | 21.5 | 22.7 | 24.1 | 71.3 | 0.16 | <0.001 | 57.3 | 53.3 | 64.3 | 83.9 | 0.13 | <0.001 |
| Difficulty in borrowing money (%) | 28.5 | 17.2 | 12.8 | 25.3 | -0.02 | 0.161 | 12.1 | 4.6 | 1.3 | 21.4 | 0.04 | 0.096 |
| Borrowed food or money (%) | 55 | 42.4 | 40.0 | 41 | -0.04 | <0.001 | 54.8 | 61.2 | 53.9 | 32.1 | -0.06 | <0.001 |
| Food insecure (%) | 18.5 | 2.0 | 1.5 | 8.4 | -0.09 | <0.001 | 6.4 | 4.0 | 0 | 1.8 | -0.15 | 0.007 |
| | Younger women | | | | | | Older women | | | | | |
| | n = 100 | n = 100 | n = 99 | n = 91 | | | n = 100 | n = 98 | n = 96 | n = 87 | | |
| Have earnings in past month (%) | 35.0 | 34.0 | 21.2 | 81.3 | 0.11 | <0.001 | 7.0 | 10.2 | 4.2 | 47.1 | 0.19 | <0.001 |
| Any savings in past month (%) | 29.0 | 28.0 | 25.25 | 80.22 | 0.14 | <0.001 | 5.0 | 8.16 | 6.3 | 46.0 | 0.22 | <0.001 |
| Any savings at all (%) | 70.0 | 69.0 | 81.82 | 100 | 0.16 | <0.001 | 33.0 | 43.9 | 53.1 | 93.1 | 0.21 | <0.001 |
| Work stress score (high = more effort) | 9.6 | 8.7 | 8.0 | 11.0 | 0.04 | 0.04 | 8.0 | 8.0 | 7.6 | 9.0 | 0.04 | 0.001 |
| Unemployment stress (high = more stress) | 9.8 | 9.5 | 9.2 | 10.0 | 0.01 | 0.654 | 9.6 | 9.2 | 8.9 | 10.0 | 0.02 | 0.151 |
| Work shame (high = more shame) | 9.3 | 8.7 | 8.4 | 8.8 | -0.03 | 0.004 | 8.9 | 8.3 | 8.1 | 7.6 | -0.07 | <0.001 |
| Done anything to earn money past 3 months (%) | 35.0 | 34.0 | 31.3 | 87.9 | 0.14 | <0.001 | 8.0 | 11.22 | 16.67 | 54.0 | 0.19 | <0.001 |
| Difficulty in borrowing money (%) | 22.0 | 16.0 | 11.1 | 16.5 | -0.04 | 0.125 | 35 | 18.37 | 14.58 | 34.5 | -0.01 | 0.641 |
| Borrowed food or money (%) | 53.0 | 40.0 | 40.4 | 31.9 | -0.06 | 0.001 | 57.0 | 44.9 | 39.6 | 50.6 | -0.02 | 0.198 |
| Food insecure (%) | 11.0 | 1.0 | 1.0 | 3.3 | -0.12 | 0.013 | 26.0 | 3.06 | 2.1 | 13.8 | -0.07 | 0.011 |

and/or sexual violence at the hands of their husbands while 18.2% of men disclosed having perpetrated such violence against their wives. Baseline results show that women were significantly less likely to have engaged in an activity to earn income in the last 3 months than men.

### Impacts on IPV and income

**Earnings and savings among women and men.** The quantitative results showed an improvement in earnings and savings of young married women and their families. Table 3 shows men and women's reported earnings and savings over time. There was a threefold increase in the proportion of women who reported having had earnings in the past month from 21% at baseline to 64.6% at 18 months (β = 0.14, p-value<0.001), and this was change was significant. Consistently more younger women reported having savings or earnings in past months than older women. Significantly more women reported having savings in the past month from 17% at baseline to 63.5% at 18 months (β = 0.17, p-value<0.001).

Likewise, the proportion of men who reported having earned money in the past month significantly increased from 52.9% at baseline to 75% at 18 months (β = 0.12, p-value<0.001).

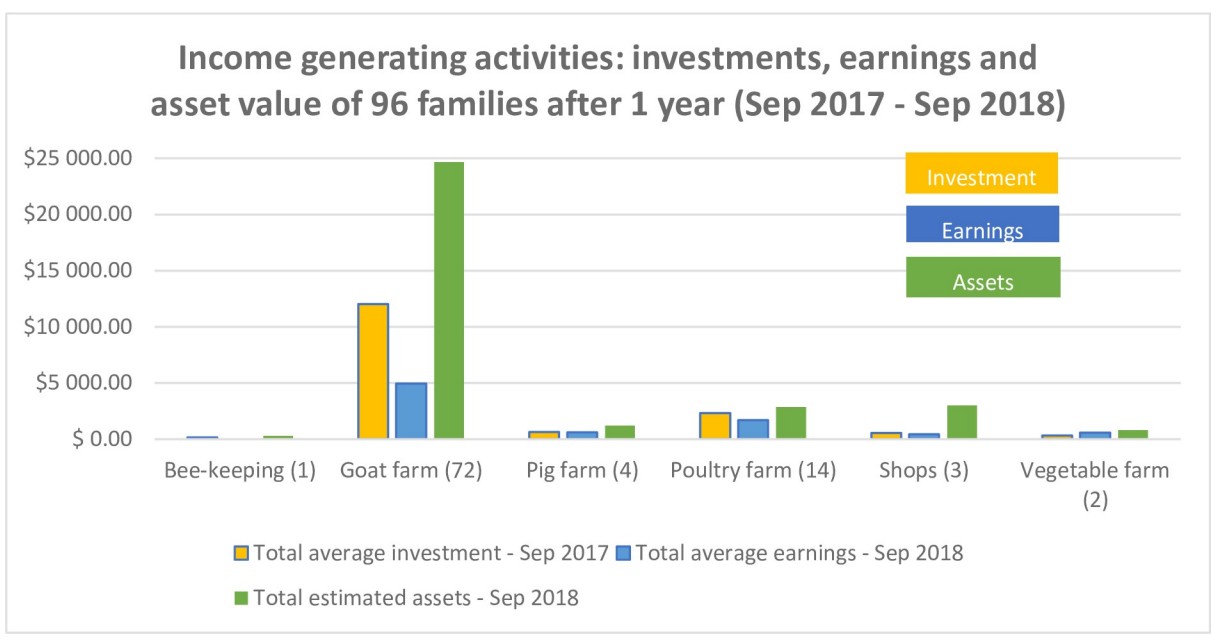

**Fig 2. Income generating activities: Investments, earnings and value assets over 1 year.**

The proportion of men who reported having had more savings in the past month increased by two-fold from 27.4% at baseline to 50.9% at 18 months (β = 0.15, p-value<0.001).

Monitoring data from the implementation of the IGAs supported the evidence that engaging in various activities to generate income had a positive impact on women's and men's earnings and saving over time. Fig 2 indicates that the IGA component of the Sammanit Jeevan programme was sustained to the end by 96 of the 100 families who were enrolled in the study. Most families (72) chose goat farming, but others chose poultry (14), pigs (4) and vegetable farming (2), shop-keeping (3) and bee-keeping (1). A year after commencement of IGAs, around September 2017, families had increased profits and asset value: overall profits ($8300) were half of the initial investment ($16040) and there was a twofold increase in asset value (estimated at $32860). Goat farming was the most profitable followed by poultry farming. However, one family reported having had to restock their poultry as the brood they had acquired at start-up was subsequently mauled by a wild animal.

Qualitative research corroborated the increases in the earnings reported in quantitative data and monitoring data. In addition to the $150 worth of start-up support, many families made in-kind contributions and sometimes supplemented from their first earnings. Women also reported having added at least one more IGA over time including gardening, breeding chickens and rearing livestock for subsistence or selling. These efforts increased their food supply and savings as they bought only those food stuffs they did not grow themselves as 18-year-old mother of one child, Sunanda, indicated: "*I buy chickens and sell them after they multiply. In the same way, I rear goats. Only today, I have bought a new goat. There's surplus grain in the house that I sell. Those are the things that I have been doing. . . These have been quite successful. We have been able to buy what we want. And we have been able to promptly sell what we want to sell. We have been making enough money to meet our needs*". For other families, reliance on their farming produce rather than buy food in the market freed them up to prioritise their children's education as Triveni suggested "*we have to spend a lot of money on education. . . as far as food is concerned we eat what we grow in our farms. . .These days, I feel that I should eat what is grown and reared in the house instead of buying things from the market*".

The reported improvements in earnings and savings can also be attributed to multiple factors including technical knowledge and skills to run IGAs. The realisation that women who are daughters-in-law can run IGAs, contrary to the prevalent gender norms that allocate household work to women and paid work to men, also empowered young women as Kusum, a 28-year-old young married mother of 2 children, suggested when she asserted: "*I got the courage from knowing that even a daughter-in-law can earn her living.*" At the end of the project, husbands reflected on their spending patterns and made changes: "*earlier we used to spend a lot of money unnecessarily. We used to buy and eat things from the shop. Five to one hundred Nepal Rupees might have been spent on that. Now we have realised that we should save that money*" (Brijesh, a 28-year-old father of one child). Other men also acknowledged the importance of young married women being able to earn their own income as suggesting that "*women can also work and earn their own living and be self-dependent*".

The achievements of increased earnings and savings from the IGAs were reportedly influenced by the supportive role of young married women's husbands and their in-laws often in ways that somewhat shifted division of labour in the household. Having recognized the delay in earnings around 6 months since commencing the IGAs, some husbands felt the need to get paid work to supplement their wives' income from her farming IGAs as 35-year-old Brojendra suggested "*we need to earn more money. I might have to leave my home and start working elsewhere if I need to make more money*". Other men recognized the social rewards accrued for them playing the male provider role but also acknowledged the importance of their wives' role in their IGAs as Brijesh explained: "*the people in the community say good things about me as I am able to earn money. . . that I have been a good son by earning money (but) I earn money elsewhere while my wife works hard here rearing the animals*". As such, the mothers-in-law were more involved with IGAs while their sons sought work outside the village and assisted their wives on the occasions they were came back home. Many young married women expressed appreciation for their mothers-in-law' support with their IGA efforts by taking over the household chores to enable daughters-in-law to concentrate on IGAs while some provided supplies at no cost. Fathers-in-law also reportedly provided invaluable advice to assist daughters-in-law maintain their IGAs and this was said of one father-in-law "*he looks after the (IGA) so that (the daughter-in-law) can then focus on the (other activities to maintain the IGA)*".

**Food security among women and men.** Consistent with the monitoring data, there were significant reductions in food insecurity and borrowing money or food in the past month (Table 3). Women's levels of food insecurity more than halved from an average of 18.5% of women with food insecurity at baseline to 8.4% at 18 months. There was a high reduction in food insecurity among younger women compared to older women (β = -0.12, p-value = 0.013 vs β = -0.07, p-value = 0.011). Reports of borrowing food or money also reduced from baseline to endline by 25% among women. Women's effort to find work significantly increased over time (β = 0.04, p-value<0.001) and they reported less shame about lack of work over time (β = -0.05, p-value<0.001). There no significant change in difficulty in borrowing money among both young and older women. However, there was a significant reduction in the proportion of younger women who borrowed money or food (β = -0.06, p-value = 0.001 vs β = -0.02, p-value = 0.198).

Among men, food insecurity was significantly reduced from a mean of 6.4 at baseline to 1.8 at endline. The proportion of men who reported borrowing money or food in the past month also reduced significantly (54.8% - 32.1%, p-value<0.001). Men reported significantly more effort into finding work over time (mean 10–11, p-value = 0.04), and less shame about lack of work at 18 months (8.7 vs 7.9, p-value = 0.02).

**Gender attitudes and relationships in the household.** Women reported significantly less patriarchal individual gender attitudes from a mean of 46.8 at baseline to 45.5 at 18 months (p-

value = 0.004) and held perceptions that the community had less patriarchal gender attitudes from a mean of 54.6 to 51.1 (p-value<0.001) (Table 4). There was more significant change in the older women's gender attitudes compared to the younger women (β = -0.09, p-value = 0.001 vs β = -0.04, p-value = 0.297). This positive change in gender attitudes was reflected among many young married women who consistently posited that "*men and women are the two wheels of the same chariot*" [Kunjana, a 33-year-old young married mother of 2 children]. Perceived relations with husbands had also improved (p-value = 0.003) as less control by husbands was reported (β = -0.07, p-value = <0.001). No change was reported on perceptions of relations with mothers-in-law among all women in the study. However, comparisons by women's in-law status indicated that young married women increasingly reported a reduction in the perception that the mother-in-law was cruel over time (mean 9.0–8.6, p-value = 0.035) while older married women had increased perceptions that their mothers-in-law were cruel and reduced perceptions that their mothers-in-law were kind.

In their qualitative interviews young married women reported that their mothers-in-law had become less cruel, particularly being cooperative and helpful to ensure that daughters-in-law implemented their IGAs successfully. Jharana, a 39-year-old mother explained: "*my mother-in-law used to ask me to work all the time. But now. . . she helps me in the kitchen, in the animal shed and. . . with other outdoor works. . . Things have been easy for me. She does not scold me. . . The relationship is very good*". This was confirmed by some mothers-in-law as 57-year-old Khushmati indicated: "*we do not consider our daughter-in-law to be someone else's daughter. All of us do the same work—including my sons and my daughter-in-law and even my husband and I*". While monitoring data demonstrating the involvement of young married women in household decision-making pertaining to the IGAs and household expenditure, this was not supported by the measure of women's involvement in household decision-making in the quantitative research. Qualitative research seemed to also indicate the families' increasing recognition of the daughter-in-law's role in income generation suggesting that young married women are "*the one(s) who manage the household expenses and. . . decide what amount is spent on what. . . what amount should be given to children, husband and parents-in-law as pocket money*". S Some young married women had also begun to challenge the notion of husbands as decision-makers as Kunjana explained: "*earlier, we believed that we had to do what our husbands asked us to do but now we have realised that we need not wait for them for everything. We have to respect them, but we don't have to wait for them. . .*" However, mothers-in-law in the study often did not acknowledge their daughters-in-law's role in the IGAs but spoke of the IGAs as theirs. Thus, women's lower reporting on household decision-making appeared to be driven by some of the in-laws assuming property ownership including the IGAs led by their daughters-in-law.

The proportion of women reporting sometimes or often quarrelling with husbands increased significantly from 41.5% at baseline to 57.9% at 18 months (β = 0.03, p-value = 0.003) and a similar pattern was observed among women reporting often quarrelling with husbands (3.5% - 10.7%, β = 0.11, p-value = 0.001). There was a significant increase in the proportion of older women reporting quarrelling with husband compared to younger women (β = 0.04, p-value = 0.028 vs β = 0.01, p-value = 0.417). In qualitative research, participants used the term "*jhagada*" to refer to both discussions and quarrels they may have with other people suggesting that increase in quarrelling may also imply improved women's expression and communication in relationships with husbands. The 12 months qualitative data suggests that young married women had become "*more vocal and confident*" to express themselves towards husbands and in-laws, compared to formative research indicating that young married women often "*keep quiet*" during arguments with husbands and in-laws [16]. Improvements in relationships between young married women and their husbands coincided with women's

**Table 4. Gender attitudes, relationships and general and mental health among men and women and among young versus older women.**

| | | **Women** | | | | | **Men** | | | | | |
| --- | --- | --- | --- | --- | --- | --- | --- | --- | --- | --- | --- | --- |
| | Baseline | 6m | 12m | 18m | | | Baseline | 6m | 12m | 18m | | |
| | mean /% | mean /% | mean /% | mean /% | coef. | p-value | mean /% | mean /% | mean /% | mean /% | coef. | p-value |
| | n = 200 | n = 198 | n = 195 | n = 178 | | | n = 157 | n = 152 | n = 154 | n = 112 | | |
| **Gender attitudes and relationships** | | | | | | | | | | | | |
| Individual gender attitudes(high = patriarchal) | 46.8 | 47 | 46.9 | 45.5 | -0.07 | 0.004 | 46.1 | 47.2 | 46.2 | 41.2 | -0.26 | <0.001 |
| Community attitudes (high = patriarchal) | 54.6 | 52.1 | 49.8 | 51.1 | -0.22 | <0.001 | 49.2 | 48.6 | 47.1 | 50.7 | 0.03 | 0.235 |
| Wife's relationship with husband (high = poorer) | 15.3 | 15.2 | 14.8 | 14.9 | -0.03 | 0.003 | - | - | - | - | - | - |
| Husband's relationship with wife (high = better) | - | - | - | - | - | - | 21.4 | 20.7 | 20.7 | 24.6 | 0.13 | <0.001 |
| Mother-in-law/mother cruelty -4items (low = good) | 8.1 | 7.9 | 7.8 | 8.4 | 0.02 | 0.161 | 8.2 | 8.2 | 8 | 6.7 | 0.08 | <0.001 |
| Mother-in-law/mother kindness (high = good) | 7.1 | 7.4 | 7.5 | 6.8 | -0.01 | 0.299 | 8.9 | 8.8 | 8.8 | 9.4 | 0.02 | 0.159 |
| Woman involvement in decision making (high = more) | 13.1 | 12.8 | 13.4 | 11.3 | -0.08 | <0.001 | 11.8 | 11.3 | 12 | 11 | 0.02 | 0.008 |
| Relationship control (high = more control) | 17.5 | 17.1 | 16.6 | 16.3 | -0.07 | <0.001 | 17 | 16.7 | 16.3 | 15 | 0.11 | <0.001 |
| Frequency of quarrelling (1 = sometimes/often) % | 41.5 | 47.5 | 33.3 | 57.9 | 0.03 | 0.003 | 26.8 | 21.1 | 11.0 | 38.4 | 0.02 | 0.344 |
| Partner alcohol use (%) | 55.0 | 53.0 | 54.4 | 63.5 | 0.04 | 0.041 | - | - | - | - | - | - |
| Alcohol use in past year (%) | - | - | - | - | - | - | 54.8 | 49.3 | 49.4 | 50 | 0.03 | 0.254 |
| **General and mental Health** | | | | | | | | | | | | |
| Depression score (low = good) | 16.9 | 13.7 | 11.1 | 16.5 | -0.08 | 0.085 | 9.9 | 9.9 | 7.4 | 11.2 | 0.02 | 0.645 |
| Life satisfaction (high = more dissatisfied) | 11.1 | 10.6 | 10.1 | 10.6 | -0.04 | 0.002 | 10.3 | 10.6 | 11 | 10.2 | 0.01 | 0.528 |
| Hope (high = more hopeful) | 18.5 | 17.8 | 17.8 | 17.4 | -0.06 | <0.001 | 18.3 | 18 | 17.9 | 20.1 | 0.08 | <0.001 |
| Self-reported health (good or excellent) (%) | 17 | 17.2 | 27.7 | 14 | 0.01 | 0.711 | 24.8 | 30.3 | 29.2 | 37.5 | 0.05 | 0.004 |
| Suicidal thoughts in past 4 weeks (%) | 7.5 | 6.6 | 1.5 | 12.9 | 0.03 | 0.193 | 0 | 0 | 0 | 0.9 | - | - |
| Self-reported impairment/disability (%) | 25 | 15.7 | 13.3 | 32 | 0.02 | 0.126 | 7.6 | 2.6 | 5.8 | 13.4 | 0.06 | 0.056 |
| | | **Young women** | | | | | **Older women** | | | | | |
| | n = 100 | n = 100 | n = 99 | n = 91 | | | n = 100 | n = 98 | n = 96 | n = 87 | | |
| **Gender attitudes and relationships** | | | | | | | | | | | | |
| Individual gender attitudes(high = patriarchal) | 44.4 | 45.1 | 46.1 | 43.3 | -0.04 | 0.297 | 49.3 | 48.6 | 47.7 | 48 | -0.09 | 0.001 |
| Community attitudes (high = patriarchal) | 54.4 | 52.1 | 50.3 | 51.4 | -0.19 | <0.001 | 54.8 | 52.1 | 49.4 | 51 | -0.25 | <0.001 |
| Wife's relationship with husband (high = poorer) | 15.3 | 15.4 | 14.9 | 14.1 | -0.07 | <0.001 | 15.5 | 15 | 14.6 | 15.8 | 0.01 | 0.438 |
| Mother-in-law cruelty—4 items (low = good) | 9 | 8.5 | 8.4 | 8.6 | -0.03 | 0.035 | 7.2 | 7.2 | 7.1 | 8.3 | 0.06 | 0.001 |
| Mother-in-law kind -3 items (high = good) | 7.9 | 8 | 8.3 | 8.1 | 0.02 | 0.163 | 6.3 | 6.8 | 6.7 | 5.5 | -0.04 | 0.005 |
| Woman involvement in decision making (high = more) | 13.4 | 13.3 | 13.9 | 11.9 | -0.07 | <0.001 | 12.8 | 12.3 | 13 | 10.7 | -0.09 | <0.001 |
| Relationship control (high = more control) | 17.5 | 17.3 | 16.9 | 16.1 | -0.07 | <0.001 | 17.4 | 16.9 | 16.3 | 16.6 | -0.06 | <0.001 |
| Frequency of quarrelling (1 = sometimes/often) % | 39.0 | 54.0 | 38.4 | 50.6 | 0.01 | 0.417 | 44.0 | 40.8 | 28.1 | 65.5 | 0.04 | 0.028 |
| Partner alcohol use (%) | 51.0 | 45.0 | 49.5 | 50.6 | 0.002 | 0.944 | 59.0 | 61.2 | 59.4 | 77.0 | 0.09 | 0.003 |
| **General and mental health** | | | | | | | | | | | | |
| Depression score | 16 | 13 | 11 | 14 | -0.18 | 0.002 | 18 | 15 | 12 | 19 | 0.02 | 0.739 |
| Life satisfaction (high = more dissatisfied) | 11.2 | 10.7 | 10.1 | 10 | -0.07 | <0.001 | 11 | 10.6 | 10.2 | 11.2 | -0.01 | 0.985 |
| Hope (high = more hopeful) | 19 | 18 | 18.1 | 19.1 | 0.01 | 0.605 | 18.1 | 17.5 | 17.4 | 15.6 | -0.14 | <0.001 |
| Suicidal thoughts in past 4 weeks (%) | 6 | 7 | 3 | 3.3 | -0.05 | 0.208 | 9 | 6.1 | 0 | 23 | 0.08 | 0.01 |
| Self-reported impairment/disability (%) | 15 | 9 | 5.1 | 14.3 | -0.02 | 0.514 | 35 | 22.5 | 21.8 | 50.6 | 0.05 | 0.021 |
| Self-reported health (good/excellent) (%) | 20 | 23 | 41.4 | 23.1 | 0.04 | 0.066 | 14 | 11.2 | 13.5 | 4.6 | -0.06 | 0.057 |

reports of being less harsh towards family members compared to before attending the Samma-nit Jeevan intervention as well as with reports of husband and in-laws exercising fewer restrictions on young married women's interactions, mobility and performance of certain traditional practices. Qualitative accounts from young married women concurred as 18-year-old mother Sunanda described: "*our relationship is much better these days. We talk to each other properly. Earlier, he used to shout at me for using Facebook. These days, he 'likes' my photos on social media*".

Among men, individual gender attitudes were less patriarchal over 18 months (mean 46.1–41.2, p-value<0001). Perceptions that men had better relations with wives significantly increased (mean 21.4–24.6, p-value<0.001) as men also reported less controlling behaviour towards wives over time (mean 17–15, p-value<0.001). Men perceived their mothers were significantly less cruel towards their wives over time in the 4-item scale (8.2–6.7, p-value<0.001). There was no significant increase in perceived mother kindness towards men's wives. Men also reported less involvement in household decision-making, and the difference over time was statistically significant (mean 11.8–11.0, p-value = 0.008).

There was non-significant increase in the proportion of men reporting quarrelling with spouse (26.8% to 38.4%, p-value = 0.334). Women reported that the extent to which their husbands drank alcohol had significantly increased over time (55% to 63.5%, p-value = 0.04), and when men were asked about their alcohol drinking in the past year, they reported a slight reduction, but it was not statistically significant (54.8% - 50%, p-value = 0.254). Consistent with reports on quarrelling with husband, more older women reported that their partner drinking alcohol in past year ($\beta$ = 0.09, p-value = 0.003 vs $\beta$ = 0.002, p-value = 0.944). Qualitative research suggested a connection between husbands' alcohol drinking and verbal and physical abuse against wives. While young married women were more likely to say that husbands' heavy alcohol drinking had lessened, older women with mother-in-law status indicated that their husbands continued to drinking and experienced violent incidents. As Leena, 41-year-old mother-in-law described a situation where: "*. . .he still has not quit drinking. In the beginning of the year, the same thing happened. He fainted and felt weak. He acted like he was mad and threatened to kill me*".

**Mental and general health among women, men and younger versus older women.** Women's feelings of wellbeing had improved over time particularly among young women. There was a sustained decrease in depression from baseline to 12 months (mean 16.9–11.1), but this trend reversed slightly at 18 months (mean 16.5, p-value = 0.085) perhaps reflecting potential temporal effects influenced by life events over the course of the study. Comparisons among women showed that the reduction in depression was statistically significant among young married women (mean 16–14, p-value = 0.002) while the decrease in the mean was not maintained among older women (mean 18–19, p-value = 0.739). A similar pattern was observed among men with a gradual decrease from baseline to 12 months (mean 9.9–7.4) and an increase at the endline (mean 11.2, p-value = 0.645). Women were also more likely to report having had suicidal thoughts in the past 4 weeks than men, and while there was a steady decrease from 6 to 12 months (7.5% - 1.5%), this was not sustained at 18 months although this increase was not significant (12.9%, p-value = 0.193). This may have been due to some temporal effects at the time of data collection. Comparisons also indicated a significant increase in suicidal thoughts in the past 4 weeks among older women over time (9% - 23%, p-value = 0.01) while a reduction in suicidal thoughts in the past 4 weeks observed among young women was not statistically significant (6% - 3.3%, p-value = 0.203). The results on suicidal ideation for men were spurious.

Life dissatisfaction also significantly reduced among women in general at the end of the study compared to men. However, the change was significant among younger women. Over

**Table 5. Reported past year IPV experienced by women and perpetrated by men over time.**

| | Women | | | | | Men | | | | |
|---|---|---|---|---|---|---|---|---|---|---|
| | Baseline | 12m | 18m | | | Baseline | 12m | 18m | | |
| | n = 200 | n = 198 | n = 178 | | | n = 157 | n = 155 | n = 112 | | |
| | % | % | % | coef. | p-value | % | % | % | coef. | p-value |
| Physical IPV in past year | 8.0 | 5.6 | 6.7 | -0.02 | 0.472 | 4.6 | 4.5 | 2.8 | -0.03 | 0.460 |
| Physical IPV in past year (2+ acts) | 5.5 | 2.5 | 6.1 | 0.01 | 0.863 | 2.6 | 3.2 | 0.9 | -0.04 | 0.398 |
| Sexual IPV in past year | 5.0 | 2.0 | 8.5 | 0.04 | 0.143 | 0.7 | 0.7 | 5.5 | 0.17 | 0.012 |
| Physical or sexual IPV in past year | 11.1 | 6.6 | 12.8 | 0.01 | 0.668 | 4.6 | 4.5 | 8.3 | 0.04 | 0.207 |
| Severe physical or sexual IPV (> 2 acts) | 8.5 | 3.5 | 11.6 | 0.02 | 0.321 | 2.6 | 3.2 | 6.4 | 0.06 | 0.122 |
| Emotional IPV in past year | 17.1 | 4.6 | 25.6 | 0.04 | 0.026 | 4.5 | 3.9 | 10.0 | 0.06 | 0.067 |
| Any IPV in past year | 20.1 | 9.1 | 28.7 | 0.04 | 0.039 | 13.9 | 9.6 | 16.0 | 0.05 | 0.069 |
| | Young women | | | | | Older women | | | | |
| | n = 100 | n = 100 | n = 91 | | | n = 100 | n = 96 | n = 87 | | |
| Physical IPV in past year | 10.0 | 8.0 | 4.4 | -0.08 | 0.077 | 6.1 | 3.1 | 9.6 | 0.03 | 0.404 |
| Physical IPV (2+ acts) in past year | 7.0 | 3.0 | 4.4 | -0.04 | 0.364 | 4.0 | 2.0 | 8.2 | 0.05 | 0.239 |
| Sexual IPV in past year | 4.0 | 3.0 | 11.0 | 0.09 | 0.041 | 6.1 | 1.0 | 5.5 | -0.01 | 0.758 |
| Physical or sexual IPV in past year | 13.0 | 10.0 | 12.1 | -0.01 | 0.763 | 9.1 | 3.1 | 13.7 | 0.03 | 0.347 |
| Severe physical or sexual IPV (> 2 acts) | 10.0 | 5.0 | 12.1 | 0.02 | 0.601 | 7.1 | 2.0 | 11.0 | 0.03 | 0.381 |
| Emotional IPV in past year | 20.0 | 7.0 | 25.3 | 0.03 | 0.256 | 14.1 | 2.0 | 26.0 | 0.06 | 0.044 |
| Any IPV in past year | 23.0 | 14.0 | 28.6 | 0.03 | 0.294 | 17.2 | 4.1 | 28.8 | 0.05 | 0.073 |

time, men were significantly more likely to report good to excellent health (24.8% - 37.5%, p-value = 0.004) compared to women (17% - 14%, p-value = 0.711). However, there was also a marginal increase in the proportion of young married women self-reporting good/excellent health that approached statistical significance (20% - 23.1%, p-value = 0.066) while good to excellent health reports had decreased among older women (14% - 4%, p-value = 0.057). Self-reported impairment increased over time among both women (25% - 32%, p-value = 0.126) and men (7.6% - 13.4%, p-value = 0.056). Older women were significantly more likely to report impairment over time (35%– 50.6%, p-value = 0.02) but there was a slight decreased in reported impairment among young women (15% - 14.3%, p-value = 0.514).

**IPV experiences of women and perpetration by men.** There were mixed results on the impact of Sammanit Jeevan on past year intimate partner violence victimisation and perpetration over 18 months (Table 5). Overall, women's experiences of any emotional, physical or sexual IPV in the past year significantly increased from 20% at baseline to 28.7% at 18 months (β = 0.04, p-value = 0.039) and men's perpetration of IPV had also increased (13.9% - 16.0%, β = 0.05, p-value = 0.069). Women reported a reduction in past year physical IPV, but this was not statistically significant (8.0% - 6.7%, β = -002, p-value = 0.472), similarly with men's reported perpetration of past year physical IPV (4.6% - 2.8%, β = -0.03, p-value = 0.460). Women's reports of experiencing severe physical abuse by husbands in the past year had slightly increased over time (5.5% - 6.2%, β = 0.01, p-value = 0.863) but had decreased among men (2.6% - 0.9%, β = -0.04, p-value = 0.398). Increases in women reporting past year sexual IPV was also observed (5.0% - 8.5%, β = 0.04, p-value = 0.143), while reporting of past year sexual IPV perpetration was significantly higher among men over 18 months (0.7% - 5.5%, β = 0.17, p-value = 0.012). The results also showed that the extent of physical/sexual IPV in the past year increased from 11.1% at baseline to 12.8% among women at 18 months (β = 0.01, p-value = 0.668) and from 4.6% to 8.3% among men (β = 0.04, p-value = 0.207). The reporting of severe physical/sexual IPV had also increased among both women (8.5% - 11.6%, β = 0.02, p-value =

0.321) and men (2.6% - 6.4%, β = 0.06, p-value = 0.122). Women also reported a significant increase in emotional IPV experiences despite a drop at 12 months (17.1% - 26%, β = 0.04, p-value = 0.026), and men perpetration of emotional IPV also increased among men but this was not significant (4.5% - 10.0%, β = 0.06, p-value = 0.067).

On the other hand, the results showed that among the women there were difference between young women and older ones (their mothers-in-law). Overall, experiences of past year emotional, physical or sexual IPV had increased among both young (23.0% - 286%, β = 0.03, p-value = 0.294) and older women (17.2% - 28.8%, β = 0.05, p-value = 0.073), but this increase approached conventional statistical significance among older women. However, young women had a reduction in their experiences of physical IPV at the hands of their husbands in the past year (10% - 4.4%, β = -0.08, p-value = 0.077) also approaching conventional statistical significance, while older women reported a slight increase in experiences of physical IPV (6% - 9.6%, β = 0.03, p-value = 0.404). Experiences of past year severe acts of physical IPV had also decreased among young women (7.0% - 4.4%, β = -0.04, p-value = 0.364) but had increased among older women (4.0% - 8.2%, β = 0.05, p-value = 0.239). However, there were significant increases in experiences of sexual IPV among younger women in the past year (4.0% - 11.0%, β = 0.09, p-value = 0.041) and no change in experiences of sexual IPV among older women (6.1% - 5.5%, β = -0.01, p-value = 0.758). There was also a slight reduction in past year physical or sexual IPV among young women (13.0% - 12.1%, β = -0.01, p-value = 0.763) albeit not significant and this had increased among older women (9.1% - 13.7%, β = 0.03, p-value = 0.347). However, severe physical or sexual IPV had increased among both younger (10.0% - 12.1%, β = 0.02, p-value = 0.601) and older women (7.1% - 11.0%, β = 0.03, p-value = 0.381). Older women experienced a bigger increase in emotional abuse by intimate partners in the past year (14.1% - 26.0%, β = 0.06, p-value = 0.044) than younger women (20.0% - 25.3%, β = 0.03, p-value = 0.256).

In the context of mixed quantitative results on IPV, qualitative research revealed that it was more common for women to report having experienced emotional and physical abuse by their husbands compared to sexual violence, though some young women complained about the sexual expectations of their husbands in a context where women were burdened with a heavy workload which combined household chores and their income generating activities. Overall, lower reporting of IPV among women in the study area at baseline was linked to reported social expectations on young married women to be silent about challenging experiences in their marital homes, owing to the prevalent conservative gender norms and attitudes at the community level [16]. Over a year, the few women who had reported emotional and physical abuse by their husbands at baseline reported changes in their husbands' violent behaviour. This was more applicable to young married women's experiences of physical intimate partner violence. Triveni, a 36-year-old and mother of 3 children, had been emotionally and physically abused by her husband, year since the start of the project reported: "*after having attended the programme, my husband has not fought with me and he has not even hit me*". While the frequency of quarrelling seemingly increased over time, this was testament to the learned communication skills which prompted women to express themselves.

## Discussion

Our findings provide considerable support for the Sammanit Jeevan whole family approach to changing inequitable gender norms and economically empowering young women, and some mixed results in the change patterns on intimate partner violence owing to measurement error identified in the analysis. Overall, the intervention has worked well for the young married women who were the main beneficiaries for which the family intervention was designed. A

similar intervention that followed the three-pronged family approach also indicates significant reductions in IPV, changes in gender attitudes and improved economic conditions of young married women that also impacted the entire family, thus lending support that this type of intervention can work well [40].

We have shown the intervention was successful in improving domestic relations of young married women, their husbands and in-laws. This is supported by both the qualitative and quantitative data indicating that women held less patriarchal gender attitudes overall and more significantly among older women; that both younger and older women believed the community gender attitudes had improved; that the controlling behaviour by husbands had reportedly lessened; while young married women had perceptions of improved spousal and mothers-in-law relations. The qualitative evaluation supported these findings suggesting that after participating in the programme young married women had gained the confidence to express themselves during arguments with husbands and in-laws and that mothers-in-law and husbands also played a supportive role in ensuring that the young women implemented their IGAs effectively. An evaluation of a similar intervention in Tajikistan mirrors these findings [40]. While quarrelling may be perceived to be a precursor to violence, the term used to measure quarrelling in the Nepali language, *jhagada*, represents any range of vigour with which one may engage in an argument, from a vigorous conversation to raise a point to an argument [16]. The qualitative findings also indicate that compared to before participation in the intervention some young married women began to challenge unequal gender norms and practices in the family, which to an extent contributed to some arguments they began having with family members but may not have resulted in women experiencing IPV.

The intervention also strengthened family livelihoods, and in particular young married women's lives, and this is an important finding among families that had relative poverty and were highly dependent on male labour migration outside Nepal. Both the quantitative and qualitative findings show that young married women's earnings and savings more than doubled over time, albeit slowly due to the nature and challenges of the income generating activities chosen by many of their families. These findings are also supported by the significant reductions in food insecurity reported by both younger and older women alike. Borrowing of food or money because they needed food was significantly reduced for young married women as well. Husbands reporting similar livelihoods improvements as seen with women was an expected spin-off as the intervention emphasises the contributions of the family in establishing and sustaining the IGAs. The qualitative data suggests that young married women often partnered with their mothers-in-law on their IGAs as men increasingly put more efforts to finding paid work outside the community to contribute to the household income. This also reflects the deeply entrenched practice of mobility and migration in their community and their association with notions of masculinity in Nepal [46, 47]. Men's reports of less work shame also suggest that the collective contribution to IGAs at the family level works to reduce the pressure men would otherwise feel in a society that emphasises men as financial providers for family income.

The evaluation also shows notable changes in men's gender attitudes. Men held less patriarchal gender attitudes, were less less controlling of their wives, and reported improved relations with their wives. This indicates a considerable shift in some of the gendered attitudes held by men after participating in the intervention in this study, and confirms the need for programmes to work with men as allies to reduce men's perpetration of intimate partner violence in some contexts [48]. This family model may have been an appropriate platform to test the potential for men to adopt more progressive gender attitudes. This evaluation also showed differences in how wives and husbands perceived husbands' alcohol drinking over time, but this is more significantly reported by older women. The qualitative findings from older women

also suggested that their husbands may have resisted lifestyle change specifically on the extent of men's alcohol drinking. However young married women's qualitative accounts of the support received from the family indicated some fathers-in-law were supportive of IGAs though mothers-in-law were their more consistent supporters. This is plausible considering older men were the most senior, less challenged and less work burdened members of the family, than the older women and male children, and were less likely to be actively involved in supporting their daughters-in-law with the IGAs.

Unlike the consistent reductions in intimate partner violence observed in the Zindagii Shoista evaluation in Tajikistan that evaluated a similar intervention to this one [40], our evaluation showed non-linear trends in the intimate partner violence data among both women and men, and young married women and older women in this study. Baseline and midline IPV data were prone to measurement error due to interviewer effects and we worked harder to reduce the measurement error at 18 months. Our efforts to reduce measurement error are demonstrable mostly in the physical violence results at 18 months particularly among women. This reduction of physical IPV among young married women is probably important though not significant, in part due to the smaller sample size. Most likely the reason sexual IPV is much higher at 18 months is due to underreporting at baseline and midline [28]. In comparison, a bigger What Works study that took place in Nepal around the same time as our study had much higher levels of IPV reporting: of the 1800 women surveyed from 36 clusters in three districts, 15.7% had reported physical violence, 18.1% ($N = 325$) sexual violence, and a quarter of them, 25.3% ($N = 455$) reported physical and/or sexual violence [49].

In general the increase reporting of IPV among younger and older women overall may also be indicative of awareness raising attributable to participation in the intervention considering the culture of silence and stigmatisation of domestic violence observed in the community during the formative stage of the OCOF project [16] and our efforts to reduce interviewer effects. Though not significant, the reductions in young married women's reporting of physical intimate partner violence is probably indicative of the impact of economic empowerment programmes on women's position, social confidence and increased bargaining power [50, 51].

This evaluation also found negative mental health outcomes for older women participating in the study. The increase in suicidality among older women reflect the complications reflected when investigating a set of outcomes at a family level, and could be attributable to multiple factors, including perceptions that their lives had not improved including the lack of improvement in relationships with husbands and their own mothers-in-laws. This may also be attributable to measurement errors considering the high extent to which suicide is stigmatised, and suicidality may have been previously underreported. However, suicidality among women in Nepal is associated with IPV, the husband's alcoholism, the culture of silence around IPV, early age of marriage, depression, and dependence on husbands for financial security [52]. While the intervention was designed to address young married women's plight in the matrimonial home, Sammanit Jeevan may have addressed some of these variables for young married women, but perhaps not as comprehensibly as needed for older women, who for instance, had probably had prolonged exposure to IPV, husbands' problematic alcohol drinking, and the mental health effects.

Older women also reported perceptions that relationships with their own mothers-in-law had worsened over time, and these are much older women more likely to be custodians of patriarchal family values [16], but were not reached by the intervention. Participation in the programme could have contributed to the study's older women's own recall of past exposure and introspection about current attitudes and behaviour towards daughters-in-law and these findings reflect further psychosocial care needs of older women beyond the study. The findings also suggest older women may also have had difficulties accessing psychosocial care services at the district level, however, the slight increase in depression and life dissatisfaction was not

significant. In comparison, young married women had improved mental and other health outcomes. Improvements in health outcomes have been observed in other studies though reductions in depression was more likely among men [41, 53], and improved life satisfaction among women [41]. Our intervention is among the few, apart from the Tajik model [40], to show reduced levels of depression among young women. This may be an effect of multiple factors including the harmonious family environment afforded through the involvement of their mothers-in-law and husbands and may be reflective of the empowerment received through the interventions. However, the negative impacts on older women's mental health require further research to understand, including how to ensure the positive impacts reverberate to the older generations in the family structure.

## Limitations

The study has a few weaknesses. These include a lack of a control arm which prevented comparisons of the intervention effects across families, but we are confident about the study findings as we collected data at three time points and have extensive qualitative research to ensure with triangulation of the quantitative study results, though we present a few qualitative accounts of the data in this manuscript. The study also has a small sample size and small geographical area for a country with a large extent of migrancy and high levels of IPV, and this may have underpowered the study and implies that prevalence estimates cannot be generalized across all married women and men in migrant communities in Nepal. Single-arm evaluations have provided valuable information on potentially suitable, acceptable, efficacious and safer interventions in the field of GBV prevention [41, 54, 55].

We also relied on self-reported data and such data is prone to multiple biases including social desirability and recall bias. We also had fewer measures of current migrant activity of husbands to understand other dimensions of its relationship with women's experiences of IPV. The attrition among men was expected due to migration for work particularly among younger men. However, our analysis showed no associations between any of the outcomes at baseline (IPV perpetration, income, savings, earnings, mental health outcomes) and attrition and we assumed that non-availability at 18 months was random and no imputation was done for missing participants at 18 months. Other IPV measures were not investigated such as domestic violence by mother-in-law and father-in-law or the relationship between older women, their husbands and their own in-laws. We attempted to ensure verification of findings by triangulating the findings with VSO team to ensure the Nepali context was represented in the interpretation of the findings.

## Lessons from the sammanit jeevan intervention

The positive impacts of this intervention are likely to be attributed to combining the gender norms intervention, whose outcomes are often seen in the long term, with the activity-based income generating activities which enables participants to practice the principles of gender equity thus potentially accelerating the desired change.

The involvement of the whole family helped to overcome the potential backlash that might have arisen had the intervention had only recruited young married women and to build a better understanding of their subjugated social position among the family members and its negative effects on the young married women and their relationships with husbands and in-laws. Previous research suggests that the family's awareness of the nature of discussions women engaged in during these economic empowerment interventions may be protective from domestic violence [56]. Our intervention sought to improve young married women's lives by recognising that their relationships as central to the gender inequalities that increase their disempowerment [57].

Drop out from the IGA was very low, in part owing to the family-centred approach as it ensured the cooperation and support to the young married women while subverting the in-laws' perceptions about the gendered domestic role of the daughter-in-law.

Participants were also provided with seed funding without expecting them to repay the initial investment, and this contributed to participant retention and ensured participants' commitment to the IGAs. Farming income generating activities usually take time to yield profits and are often confronted with challenges that delay impactful results. However, the skills and technical advice provided by BYC business assistants supported young married women to effectively manage their IGA and boosted their ability to return profits. In this intervention the real benefits were observed at least 12 months since the start of the intervention due to intermittent earnings and participants efforts to boost their IGAs and increase their productivity.

The intervention strength also came from its sound theory of change, the participatory methods used, delivery to peer groups as well as its focus on building skills in critical reflection and communication [42]. This programme was manualised thus very structured, and fidelity to the method of delivery was possible to maintain [58]. The extensive training and support of facilitators and implementers was critical for effective implementation, as was their recruitment from similar backgrounds as the participants' which enabled them to have a local understanding of social issues facing the community and were able to gain their trust [58]. The continuous support from an experienced NGO also played a significant role in IGA management and its sustainability over time.

## Conclusions

Some caution is needed in the attribution of effect in this evaluation as there was no control arm to compare the intervention impact. However, the mixed method results suggest that with adequate time and in-kind seed funding, this family-centred approach had succeeded over 18 months in enabling considerable income generation, and the intervention strengthened the position of young women in the households and reduced some of the young married women's exposure to gender-based violence in this rural community in Nepal. This programme would benefit from further research to optimise its delivery and impact.

## Acknowledgments

The One Community One Family project is a collaborative project of Voluntary Services Overseas (Nepal) and the South African Medical Research Council. The implementation of Sammanit Jeevan was led by Mani Bhadra Sharma from Bhimapokhara Youth Club, an NGO with 30 years of working in rural community development and livelihoods strengthening. Data was collected by FACTS Research and Analytics led by Prabodh Acharya. The project also received technical support from Julienne Corboz and Helen Appleton as well as the National Women's Commission Nepal. Further information about the project is available at www.whatworks.co.za.

## Author Contributions

**Conceptualization:** Nwabisa Shai, Geeta Devi Pradhan, Esnat Chirwa, Alice Kerr-Wilson, Rachel Jewkes.

**Formal analysis:** Nwabisa Shai, Ratna Shrestha, Abhina Adhikari, Esnat Chirwa.

**Funding acquisition:** Geeta Devi Pradhan.

**Investigation:** Nwabisa Shai, Ratna Shrestha, Abhina Adhikari.

**Methodology:** Nwabisa Shai, Geeta Devi Pradhan, Ratna Shrestha, Abhina Adhikari, Esnat Chirwa, Alice Kerr-Wilson.

**Project administration:** Ratna Shrestha, Abhina Adhikari.

**Software:** Nwabisa Shai, Esnat Chirwa.

**Supervision:** Geeta Devi Pradhan, Ratna Shrestha, Abhina Adhikari, Alice Kerr-Wilson.

**Validation:** Nwabisa Shai, Geeta Devi Pradhan, Ratna Shrestha, Abhina Adhikari, Esnat Chirwa, Alice Kerr-Wilson, Rachel Jewkes.

**Writing – original draft:** Nwabisa Shai, Esnat Chirwa.

**Writing – review & editing:** Nwabisa Shai, Geeta Devi Pradhan, Ratna Shrestha, Abhina Adhikari, Alice Kerr-Wilson, Rachel Jewkes.

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
