## [Decision Letter · Decision Letter 0]

23 Oct 2019

PONE-D-19-25458

“ I got courage from knowing that even a daughter-in-law can earn her living”: Evaluation of a family-centred intervention to prevent violence against women and girls in Nepal

PLOS ONE

Dear Dr. Shai,

Thank you for submitting your manuscript to PLOS ONE. After careful consideration, we feel that it has merit but does not fully meet PLOS ONE’s publication criteria as it currently stands. Therefore, we invite you to submit a revised version of the manuscript that addresses the points raised during the review process.

I find the second reviewer's points about the negative findings especially important, and would strongly advise the authors to be up front and explicit in your discussion of these findings. 

We would appreciate receiving your revised manuscript by Dec 07 2019 11:59PM. To enhance the reproducibility of your results, we recommend that if applicable you deposit your laboratory protocols in protocols.io, where a protocol can be assigned its own identifier (DOI) such that it can be cited independently in the future. For instructions see: http://journals.plos.org/plosone/s/submission-guidelines#loc-laboratory-protocols

We look forward to receiving your revised manuscript.

Kind regards,

Lindsay Stark

Academic Editor

PLOS ONE

**Journal Requirements:**

**Comments to the Author**

1. Is the manuscript technically sound, and do the data support the conclusions?

Reviewer #1: Yes

Reviewer #2: Partly

2. Has the statistical analysis been performed appropriately and rigorously? 

Reviewer #1: Yes

Reviewer #2: Yes

3. Have the authors made all data underlying the findings in their manuscript fully available?

Reviewer #1: No

Reviewer #2: No

4. Is the manuscript presented in an intelligible fashion and written in standard English?

Reviewer #1: Yes

Reviewer #2: Yes

5. Review Comments to the Author

Reviewer #1: The authors are commended on conducting an important study that captures data from multiple people within a family, which is a rare accomplishment in studies seeking to reduce violence against women. A number of revisions could be made to strengthen the manuscript.

Abstract

1. Background: I would remove the funding source (DFID What Works) to save word count to bolster other sections. "young married women" is repeated twice so removal of one reference would be preferred.

2. Results: The results are unclear as the measures are not specified. For example, frequency changes in 'women's earnings' and 'savings' is not clear until the reader reviews the measures table. Clarification of the gender attitudes and perceptions of husband's control is needed for whether the authors are referring to young women's changes or all women's, including mothers-in-law.

Background

Overall, the background is well written and it is appreciated that the authors include customary practices as well as legal statutes in the description. However, in the first paragraph, the authors write 'Nepal's IPV prevalence is at least as high as other South Asian countries' but it is unclear what the general trend is in other South Asian countries, so it is recommended to include this information.

Methods

1. VDC is used in the participant section before it is defined.

2. Further description of how the authors accounted for ethical considerations when interviewing both men and women in the same home about IPV perpetration and victimization, respectively. Typically, this is not common in IPV research due to ethical concerns and potential backlash of men. Were any additional steps taken?

3. The HFIAS is typically nine items. Description is needed as to which items were included and why a smaller number was selected.

4. Were the husband's family living with the index younger married women? It is unclear if this was an inclusion criteria.

5. A sentence should be added on literacy levels of participants to complete for signing informed consent, especially as more than half of women reported no education. Was this a written or oral consent?

Results

1. In the second paragraph of the results, it is written 'women were more likely to have been exposed to IPV in their lifetime compared to men'. Given the following sentence includes statistics, this sentence is recommended for removal or revision as it is not clear whether one is referring to victimization or perpetration.

2. An interesting finding is that older women's report of their own mother-in-law relationship gets worse is not addressed in the discussion. Insights into this change should be included.

Discussion

1. The first paragraph states support for the intervention reducing IPV, but this seems a stretch interpretation as only physical IPV among younger women was reduced, with null or increased findings for other forms of IPV, particularly for older women. Rephrasing this conclusion would be recommended.

2. The paragraph describing the increase in IPV being related to under-reporting at baseline is possible, but it does not address why IPV had gone down at the midpoint periods of data collection for a few forms of IPV. An additional alternate interpretation is whether the intervention reduced men's migratory practices for work. If increased economic wellbeing could lead to a reduction in this, it is possible men were home more to perpetrate IPV at endline. If this, or other alternate explanations seem viable, it should be included in this paragraph.

3. The sentence on alcohol use seems out of place and additional information could be added as to what parts of the intervention addressed alcohol use.

Reviewer #2: Thank you for providing me with the opportunity to review this article. The authors discuss the evaluation of three-pronged intervention to reduce violence against women and improve their economic well-being. The article is well-written and easy to follow, and the findings are important to share with the development community. However, in addition to the areas for improvement outlined below, I am concerned with the lack of attention paid to the negative outcomes observed in the findings. The authors conclude, “Our findings provide considerable support for the Sammanit Jeevan whole family approach to reducing intimate partner violence, changing inequitable gender norms and economically empowering young women”, yet several adverse effects were also observed, such as a decline in self-reported health, increase in suicide ideation for older women, increase in frequency of quarreling (as reported by women), increase in IPV for older women, increase in sexual violence perpetration by men, among several others. In addition to addressing the comments below, I would highly recommend that the authors devote considerable attention to these negative findings in the discussion section (in addition to amending the conclusion given above).

Background

1. The third paragraph does an excellent job of explicating the factors contributing to young married women’s experiences of violence as perpetrated by their mothers-in-law and how these factors are linked to gender inequality and patriarchal family structures. However, the majority of the Background is focused on this topic, which is not the primary outcome of the intervention as stated in Table 1. I believe the second paragraph could be strengthened by expanding on the mechanisms that foster IPV in this setting, rather than simply stating various characteristics found to be associated with IPV in Nepal.

2. Given that primary outcomes also include economic- and income-based indicators, the Background section should provide background on women’s labor force participation and income decision-making in this context, including barriers to labor participation and the underlying gendered power dynamics that inherently marginalize women in this area. Such discussion of this literature will make clear for the reader why the intervention includes the three primary components of gender transformative norms, economic empowerment, and IGAs support.

3. What evidence currently exists around the effectiveness of programs like Sammanit Jeevan in Nepal or elsewhere? Has any evaluation to date looked at the impact of an intervention that includes all three components? Please summarize existing evidence around these intervention activities as it relates to the target outcomes. Further and importantly, the study shows very mixed evidence on several of the outcomes, without any way to determine which components of the study might have contributed to the negative consequences (increase in suicide ideation for older women, overall increase in sexual violence for younger women, etc.).

Methods

4. How was your sample size calculated?

5. Did all invited individuals agree to participate in the study? The flowchart in Figure 1 should be expanded to include the number of invited participants as a first row.

6. “Sessions run once weekly with separate same age-sex groups which came together for discussions every third session.” Does this mean there were four groups (100 younger women, 100 older women, 78 younger men, and 79 older men), or were these groups further broken down into several sessions? Please provide more details about the implementation of the program sessions and group makeup.

7. How long was each of the three intervention phases? Between what dates was the intervention implemented? Was the intervention still ongoing during the 6-month follow-up point of data collection? Please clarify the timeline in the text and in Figure 1.

8. Were female interviewers used for female participants?

9. “Data was collected on the implementation of IGAs per family.” Was this data collected as part of the in-depth interviews? The surveys? Additional data collection? Please elaborate.

10. Given attrition, particularly among men, how was non-response addressed in the analysis? Did the authors use imputation? Was any analysis conducted to assess whether missing data was associated with outcomes of interest at earlier time points. I highly recommend you assess whether attrition is associated with these outcomes or, at the very least, discuss the implications of this in the limitations/discussion section.

Results

11. There is a Table 1 and Figure 1 in the methods section as well as the Results section. Please check the Table and Figure labels and references in text.

12. Table 1 in the Results section. I would suggest separating the summary statistics for the four main groups in the sample: younger women, older women, younger men, and older men.

13. Additionally, is it possible to report past-year IPV rather than lifetime, given that past-year is the timeframe for the primary outcomes?

14. Table 1. Am I reading correctly that 14% of women report experiencing sexual IPV and 0.7% of men report perpetrating?

15. The intervention is framed in the Background section as intended to target outcomes for young married women; however, the results for women are presented altogether (for young and older women combined). Is there a reason the authors decided to present the results in this way? All relevant outcomes and trends should be presented for the young married women’s group separately as well. For example, in Table 2, it is not clear if the increase in income for women is found in both the younger and older groups (and this distinction seemingly matters given the justification for the intervention). The same comment for the first panel in Table 3 as well.

Discussion

16. The increase in suicide ideation and self-reported impairment for the older female cohort is a worrisome finding. What do you attribute this increase to and how might this be mitigated in future iterations of the intervention?

17. There is only brief mention of lack of control group as a limitation. The authors should also discuss the self-reported nature of the outcomes, small sample size, and potentially biased attrition (as mentioned in comment #10) as limitations of this study.

18. My primary concern with the discussion section is the conclusion drawn in the first sentence: “Our findings provide considerable support for the Sammanit Jeevan whole family approach to reducing intimate partner violence, changing inequitable gender norms and economically empowering young women.” While the study did find improvement for some outcomes, several adverse effects were also observed, such as a decline in self-reported health, increase in suicide ideation for older women, increase in frequency of quarreling (as reported by women), increase in IPV for older women, increase in sexual violence perpetration by men, among several others. While increased comfort with reporting incidents of IPV may explain the increase in violence exposure, further qualitative research is needed to explore the extent to which this explanation accounts for the observed increase. Further, this likely does not explain the increase in suicide ideation and other negative outcomes for older women. The discussion section needs to more accurately discuss the potential adverse impacts of this intervention, how and which intervention activities might have contributed to these outcomes, and future implementation of the intervention might avoid these outcomes. Further, given all of this, I am not comfortable with the conclusion outlined in the first sentence of the Discussion.

Abstract

1. Please add the study setting to the abstract

2. The stats provided in parentheses are not clear- is this a range? A change from X to Y?

Grammatical edits to the text are also needed.

6. PLOS authors have the option to publish the peer review history of their article (what does this mean?). If published, this will include your full peer review and any attached files.

Reviewer #1: No

Reviewer #2: No

---

## [Author Response · Author response to Decision Letter 0]

21 Dec 2019

The reviewers' coments have been addressed and are presented on a table in the cover letter

---

## [Decision Letter · Decision Letter 1]

11 Mar 2020

PONE-D-19-25458R1

“ I got courage from knowing that even a daughter-in-law can earn her living”: mixed methods evaluation of a family-centred intervention to prevent violence against women and girls in Nepal

PLOS ONE

Dear Dr. Shai,

Thank you for submitting your manuscript to PLOS ONE. After careful consideration, we feel that it has merit but does not fully meet PLOS ONE’s publication criteria as it currently stands. Therefore, we invite you to submit a revised version of the manuscript that addresses the points raised during the review process.

I would like to note that a new reviewer was invited at this stage when one of the original reviewers was not available. The new reviewer has recommended that this paper not be put forward for publication in PLOS One, however, we would like to give the authors to respond to the feedback so that the editorial board can make a final informed decision. We thank the authors in advance for doing what they can to address the reviewer concenrs.

We would appreciate receiving your revised manuscript by Apr 25 2020 11:59PM. To enhance the reproducibility of your results, we recommend that if applicable you deposit your laboratory protocols in protocols.io, where a protocol can be assigned its own identifier (DOI) such that it can be cited independently in the future. For instructions see: http://journals.plos.org/plosone/s/submission-guidelines#loc-laboratory-protocols

We look forward to receiving your revised manuscript.

Kind regards,

Lindsay Stark

Academic Editor

PLOS ONE

Reviewers' comments:

Reviewer's Responses to Questions

**Comments to the Author**

1. If the authors have adequately addressed your comments raised in a previous round of review and you feel that this manuscript is now acceptable for publication, you may indicate that here to bypass the “Comments to the Author” section, enter your conflict of interest statement in the “Confidential to Editor” section, and submit your "Accept" recommendation.

Reviewer #2: All comments have been addressed

Reviewer #3: (No Response)

2. Is the manuscript technically sound, and do the data support the conclusions?

Reviewer #2: Yes

Reviewer #3: No

3. Has the statistical analysis been performed appropriately and rigorously? 

Reviewer #2: Yes

Reviewer #3: No

4. Have the authors made all data underlying the findings in their manuscript fully available?

Reviewer #2: Yes

Reviewer #3: Yes

5. Is the manuscript presented in an intelligible fashion and written in standard English?

Reviewer #2: Yes

Reviewer #3: Yes

6. Review Comments to the Author

Reviewer #2: The authors have done a very satisfactory job of addressing my comments and I recommend this manuscript for acceptance. However, given that PLOS ONE does not copyedit accepted manuscripts, the manuscript might benefit from one more round of editing for grammatical errors.

Reviewer #3: This is a well written manuscript that details the effectiveness of the Samaanit Jeevan intervention in Nepal. There are a few limitations that dampen my enthusiasm for this manuscript, and one specifically that needs to be addressed (A) without some type of comparison group (formal control arm, analytic control methods, sensitivity analyses), it is impossible to evaluate if the program was effective or if this study documents an artifact of the historical context where individuals and community values are changing; (B) insufficient attention to sample size, especially with diverse ethnic groups in region; and (C) intervention design has some key limitations – in addition to comparison group issue - including discussions on how other family members participated in programming. I elaborate on all points in my comments for the authors below, however sufficient detail was provided by previous reviewers and thus I make an attempt to avoid double commenting on those items.

These are:

(A) Lack of comparison group. While highlighted by the authors as a limitation, this design flaw was not addressed analytically at all. Effectiveness cannot be established without comparing findings to those who did not receive the intervention, especially in communities and contexts that are changing as rapidly as modern Nepali villages are changing. Additional work with inverse probability weighting or sensitivity analyses could have at least worked towards addressing these issues. However, as written, the basic conclusions around effectiveness lack evidence to support them.

(B) Further discussion of sample size considerations would also strengthen this manuscript. Given the different ethnicities presented in the study population and the heterogeneous nature of attitudes towards IPV, further discussion and perhaps sensitivity analyses by primary ethnic groups is warranted.

(C) The overall intervention design has a few components that are further troubling, even if the above two issues are/were addressed. For example, there is a long history in Nepal of lower-powered individuals (be it by caste, ethnicity, or gender) being manipulated by socially proximal higher powered individuals. Further discussion into how ‘other family members’ who participated with the women were selected, their backgrounds, and the nature of their input, is essential to understand the true scope and impact of this intervention.

7. PLOS authors have the option to publish the peer review history of their article (what does this mean?). If published, this will include your full peer review and any attached files.

Reviewer #2: No

Reviewer #3: No

---

## [Author Response · Author response to Decision Letter 1]

18 Mar 2020

Dear Reviewers, 

We thank you for your comments. Please find the authors' response in a SEPARATE file: "Responses to Reviewers".

Best, 

Authors

---

## [Editor Report · Decision Letter 2]

13 Apr 2020

“ I got courage from knowing that even a daughter-in-law can earn her living”: mixed methods evaluation of a family-centred intervention to prevent violence against women and girls in Nepal

PONE-D-19-25458R2

Dear Dr. Shai,

We are pleased to inform you that your manuscript has been judged scientifically suitable for publication and will be formally accepted for publication once it complies with all outstanding technical requirements.

With kind regards,

Shelina Visram, PhD, MPH, BA

Academic Editor

PLOS ONE

Additional Editor Comments (optional):

You have provided sufficient justification for your choice of study design and its usefulness in this and other contexts. More could perhaps have been done to emphasise that the findings should be treated with caution due to the lack of control group (is it not possible that any changes observed could be attributed to being married for an additonal 18 months, as opposed to taking part in the intervention?), but the study has certainly demonstrated the feasibility of this important intervention. As a result, I feel the manuscript is now ready for publication in PLOS One.
---

## [Editor Report · Acceptance letter]

4 May 2020

PONE-D-19-25458R2 

“ I got courage from knowing that even a daughter-in-law can earn her living”: mixed methods evaluation of a family-centred intervention to prevent violence against women and girls in Nepal 

Dear Dr. Shai:

I am pleased to inform you that your manuscript has been deemed suitable for publication in PLOS ONE. Congratulations! Your manuscript is now with our production department. 

With kind regards,

on behalf of

Dr. Shelina Visram 

Academic Editor

PLOS ONE